# Characterization of Anthocyanins Including Acetylated Glycosides from Highbush Blueberry (*Vaccinium corymbosum* L.) Cultivated in Korea Based on UPLC-DAD-QToF/MS and UPLC-Qtrap-MS/MS

**DOI:** 10.3390/foods14020188

**Published:** 2025-01-09

**Authors:** Ju Hyung Kim, Ryeong Ha Kwon, So Ah Kim, Hyemin Na, Jeong-Yong Cho, Heon-Woong Kim

**Affiliations:** 1Department of Agro-Food Resources, National Institute of Agricultural Sciences, Rural Development Administration, Wanju 55365, Republic of Korea; juuhkim0128@gmail.com (J.H.K.); haha08447@gmail.com (R.H.K.); soah0726@gmail.com (S.A.K.); gpals0713@gmail.com (H.N.); 2Department of Integrative Food, Bioscience and Biotechnology, Chonnam National University, Gwangju 61186, Republic of Korea

**Keywords:** blueberry, anthocyanin, acetylated glycoside, UPLC-DAD-QToF/MS, UPLC-Qtrap-MS/MS

## Abstract

In this study, anthocyanin glycosides from nine cultivars of highbush blueberries grown in Korea were characterized using UPLC-DAD-QToF/MS and UPLC-Qtrap-MS/MS. A total of twenty-two derivatives were identified, consisting of mono-glycosides and acetyl-glycosides attached to aglycones, such as cyanidin, peonidin, delphinidin, petunidin, and malvidin. Among them, seven acetylated glycosides were tentatively determined by comparing the related authentic standards and previous reports and presented mass fragmentation, in which the acetyl group remained as the form attached to the sugar without de-esterification in positive ionization mode. The mid-season cultivar ‘New Hanover’ showed the highest total anthocyanin content (1011.7 mg/100 g dry weight) with predominant malvidin and delphinidin glycosides. Particularly, the ‘Patriot’ (early season) recorded the highest proportion of acetylated glycosides (19.7%). Multivariate analysis showed a distinct separation between early and mid-seasons with Draper. Especially, delphinidin 3-*O*-galactoside (VIP = 1.94) was identified as a marker for mid-season, and malvidin 3-*O*-glucoside (VIP = 1.79) was identified as a marker for early season. These comprehensive anthocyanin profiles of Korean blueberries will serve as fundamental data for breeding superior cultivars, evaluating and developing related products as well as clinical and metabolomic research.

## 1. Introduction

Blueberries (*Vaccinium* spp.), along with bilberries, belonging to the *Vaccinium* genus of the Ericaceae family, which comprises approximately 400 species, are mainly distributed in Southeast Asia. These species are classified into highbush (*V. corymbosum* L.), lowbush (*V. angustifolium* L.), and rabbiteye (*V. ashei* L.). The blueberries grown in Korea are of the highbush type, which is characterized by cold hardiness and suitability for cultivation [1]. In general, highbush blueberries can be further classified as northern and southern depending on chilling requirements [2], and early-, mid-, and late-season cultivars by harvest time [3]. Early-season cultivars are typically harvested from late May to early June, while mid-season cultivars are harvested from mid to late June, and late-season cultivars are harvested from early to late July [4].

Blueberries are known to be rich in phenolic compounds (anthocyanins and flavonoids), carotenoids, and vitamin C [5], with closely related antioxidant [6], anticancer [7], antidiabetic [8], and cardiovascular disease prevention properties [9]. Anthocyanins are the most abundant and characteristic polyphenols in blueberries, accounting for 35–74% of the total phenolics in blueberries [10]. Blueberry anthocyanins are composed of aglycones, such as delphinidin, cyanidin, petunidin, peonidin, and malvidin, with sugar moieties (glucose, galactose, and arabinose) attached. Among them, malvidin (37–52%) and delphinidin (27–31%) glycosides are the most commonly identified [11,12]. Delphinidin, petunidin, and malvidin derivatives, identified as major components, are closely associated with the blue and purple colors [13]. Additionally, blueberry anthocyanins have antioxidant and anti-inflammatory activities, contributing to various health benefits (Appendix A). Acetylated anthocyanins, which are present in small amounts in blueberries, have a structure in which an acetyl group is acylated at the 6″-OH position of glucose or galactose. These compounds are reported to contribute to the intensity and stability of pigments in foods [14] and exhibit antioxidant and *β*-glucosidase effects [15,16], suggesting applications in functional foods and pharmaceuticals [17].

Recently, studies on the identification of individual anthocyanins have been actively conducted in blueberries using mass spectrometry (MS). A total of 25 anthocyanin glycosides were detected and fragmented through positive ionization mode of ESI-ToF-MS from southern highbush blueberries in the United States [18]. The anthocyanin composition and content in blueberries were distributed differently depending on their cultivar and cultivated condition [19,20]. Although 74 cultivars grown in China were characterized by Chai et al. [21], only total content was provided, without detailed individual components for acetylated glycosides, which account for 0–42% of total anthocyanins. The structural elucidation of acetylated glycosides has rarely been performed through NMR [22] and LC-MS [23,24] analyses. Since the overall quantification of blueberry anthocyanins primarily focused on major groups, such as non-acetylated glycosides [25], these detailed profiles are still limited.

The objective of this study was to rapidly and precisely identify and quantify individual anthocyanins, including acetylated glycosides from nine highbush blueberry cultivars (Reka, Hannah’s Choice, Spartan, Draper, Suziblue, Legacy, New Hanover, Farthing, and Patriot) grown in Korea, based on ultra-performance liquid chromatography–diode array detection–quadrupole time-of-flight mass spectrometry (UPLC-DAD-QToF/MS) and ultra-performance liquid chromatography–hybrid triple Q-linear ion trap mass spectrometry (UPLC-Qtrap-MS/MS) with relative quantification and external quantification methods. In addition, multivariate analysis visualized differences between cultivars and identified discriminant markers critical for cultivar differentiation.

## 2. Materials and Methods

### 2.1. Plant Materials

Blueberries (*Vaccinium* spp.) were obtained from a farm in Anseong-si, Gyeonggi-do (latitude/longitude: 37°0′4.7″ N/127°12′35.7″ E), in June 2023. The samples were collected with the permission of the Rural Development Administration (RDA, Republic of Korea) and identified by the author (Heon-Woong Kim) of this article. Detailed information about each cultivar is provided in Table 1. The experimental research, including the collection of plant material, complied with relevant institutional, national, and international guidelines and legislation. All samples were lyophilized, crushed with a grinding machine, and sieved through a 30-mesh sieve.

### 2.2. Chemicals

Cyanidin 3-*O*-arabinoside, cyanidin 3-*O*-galactoside, cyanidin 3-*O*-glucoside, delphinidin 3-*O*-glucoside, malvidin 3-*O*-galactoside, malvidin 3-*O*-glucoside, peonidin 3-*O*-arabinoside, peonidin 3-*O*-galactoside, peonidin 3-*O*-glucoside, petunidin 3-*O*-glucoside, and delphinidin 3,5-di-*O*-glucoside (delphin) were purchased from Extrasynthese (Genay Cedex, France). Petunidin 3-*O*-galactoside was provided by Chemfaces (Wuhan, China). LC-MS-grade acetonitrile and water were obtained from Thermo Fisher Scientific (Fair Lawn, NJ, USA), and formic acid from Junsei Chemical Co. (Tokyo, Japan).

### 2.3. Extraction and Sample Preparation

For UPLC-DAD-QToF/MS and UPLC-QTrap-MS/MS analyses, powdered blueberry (0.5 g) was extracted with 10 mL of 5% formic acid in water using an orbital shaker (for 40 min at 200 rpm) and centrifuged at 2016× *g* and 10 °C for 15 min (LABOGENE 1580R, Bio-Medical Science Co., Seoul, Republic of Korea). The supernatant was filtered through a 0.45 μm syringe filter and purified using a HyperSep Retain-PEP solid-phase extraction (SPE) cartridge (Thermo Scientific, Bellefonte, PA, USA). In order to remove undesirable components and improve separation and detection in UPLC-MS, the SPE was performed, and the process was as follows: The cartridge was conditioned with methanol (4 mL) and water (8 mL). The samples (0.5 mL) and internal standard solution (delphin, 100 ppm; 1 mL) were loaded into the cartridge and washed with water (6 mL). Finally, the loaded samples were eluted with 1% formic acid in methanol (12 mL) and concentrated using N_2_ gas, then re-dissolved with 5% formic acid in water (0.5 mL).

### 2.4. Anthocyanin Identification and Quantification

The tests of anthocyanin derivatives in different blueberry cultivars were comprehensively performed using an UPLC system coupled with a diode array detector (DAD; ACQUITY UPLC™ system, Waters Co., Milford, MA, USA) and a QToF-MS (Xevo G2-S QToF, Waters MS Technologies, Manchester, UK). Chromatographic conditions were as follows: CORTECS UPLC T3 column (2.1 × 150 mm, 1.6 μm, Waters, Wexford, Ireland) and CORTECS UPLC T3 VanGuard™ pre-column (2.1 × 50 mm, 1.6 μm, Waters), representative detection wavelength: cyanidin and peonidin, 515 nm; delphinidin, 520 nm; petunidin and malvidin, 525 nm; flow rate, 0.3 mL/min; column oven temperature, 30 °C; sample injection volume, 1 μL; mobile phase, 5% formic acid in water (A) and 5% formic acid in water/acetonitrile (1:1, *v*/*v*) (B). Elution gradient conditions were as follows: 0 min, 10% B; 28 min, 50% B; 33–38 min, 90% B; 43–50 min, 10% B. Mass spectra were simultaneously scanned with the range of 50–1200 *m*/*z* in positive ionization mode using an electrospray ionization (+ESI) source, and the parameters used were as follows: capillary voltage, 3500 V; sampling cone voltage, 40 V; extraction cone, 4.0 V; ion source, 120 °C; de-solvation temperature, 500 °C; de-solvation N_2_ gas flow, 1020 L/h. All standard compounds were externally quantified based on multiple reaction monitoring (MRM) mode using UPLC-QTrap-MS/MS (SCIEX QTrap 4500, SCIEX CO.), whereas other anthocyanins were quantified internally using QToF/MS (Xevo G2-S QToF, Waters MS Technologies, Manchester, UK). The optimizing MRM conditions are shown in Table 2. Through preliminary experiments, delphin, which does not overlap with sample peaks and has a similar structure to blueberry anthocyanins, was selected as the ISTD, and the relative quantification was calculated by comparing the relative peak areas of the compounds (based on major fragment ions) and ISTD on a 1:1 basis without considering the relative response factor. The external quantification was performed in MRM mode with 11 selected standards. The contents of anthocyanin derivatives were expressed in mg/100 g DW (dry weight). The triplicate results are expressed as mean ± standard deviation. One-way ANOVA was performed with SPSS (version 28.0, SPSS institute; Chicago, IL, USA) to determine a significant difference between individual averages using Duncan’s multiple range test (*p* < 0.05). To quantify anthocyanins from blueberry, linearity was determined by analyzing standard solutions at concentrations of 0.01, 0.05, 0.1, 0.5, 1, 2, and 5 µg/mL using UPLC-QTrap-MS/MS, with each concentration analyzed in triplicate. The calibration curves were constructed by plotting the peak area against the concentration of the corresponding standards using least-square linear regression (Table 2). LOD and LOQ for used standards were calculated using the equations LOD = 3.3 × SD/δ and LOQ = 10 × SD/δ, respectively, where SD represents the standard deviation of the response (y-intercept), and δ is the slope of the calibration curve. Intra- and inter-day precisions were evaluated by analyzing standard mixtures (0.5 μg/mL, *n* = 6) on a single day and at 6 days. The variations were expressed as the relative standard deviation (RSD) of the replicates (Table 3).

### 2.5. Muliple Anlysis for Anthocyanins

A multivariate analysis was conducted using SIMCA (version 16, Satorius Stedim Data Analysis AB, Umeå, Sweden). Principal component analysis (PCA) and orthogonal partial least-squares discriminant analysis (OPLS-DA) were applied for statistical comparison and marker identification of nine blueberry cultivars. The scaling method was performed using Pareto (par). Chemical markers that distinguish the two species were obtained by combining S-plots and variable importance in projection (VIP) plots. Chemical markers were selected with VIP > 1 and *p*-value < 0.05 to indicate their influence on the classification. To evaluate the reliability and predictability of the OPLS-DA model, 7-fold cross-validation (CV), 200 random permutation tests, and receiver operating characteristic (ROC) curve models were performed. Additionally, hierarchical clustering analysis (HCA) with a heat map was constructed using the Euclidean distance method using MetaboAnalyst online analysis software 6.0 (http://www.metaboanalyst.ca, accessed on 1 December 2024) for clustering and visualization of the chemical markers between early and mid-seasons.

## 3. Results and Discussion

### 3.1. Identification of 22 Anthocyanin Glycosides from Blueberry Cultivars

In blueberries, anthocyanins are primarily identified as glycosides of cyanidin (*m*/*z* 287), peonidin (*m*/*z* 301), delphinidin (*m*/*z* 303), petunidin (*m*/*z* 317), and malvidin (*m*/*z* 331). These compounds consist of a sugar moiety (galactose, glucose, or arabinose) attached to the 3-OH position of the aglycone and acylated with acetic acid at the 6″-OH position of the sugar (Table 4 and Figure 1).

A total of twenty-two derivatives were tentatively identified as galactoside (5), glucoside (5), arabinoside (5), acetyl-galactoside (2), and acetyl-glucoside (5) based on cyanidin, peonidin, delphinidin, petunidin, and malvidin aglycones and confirmed with the presence of galactose (162 Da), glucose (162 Da), arabinose (132 Da), acetyl-galactose (42 + 162 Da), and acetyl-glucose (42 + 162 Da) moieties from the whole structure. Particularly, in positive ionization mode, the acetylated glycosides were fragmented, in which the acetyl group remained as the form attached to the sugar without de-esterification [26,27]. To further characterize these compounds, the previously reported retention times (Rt, min) [28], fragmentation patterns [29], ultraviolet spectra (UV) [30], and other sources [31] supported the isolation and identification of individual anthocyanins with UPLC-DAD-QToF/MS experimental data (Table 4 and Figure 2, Figure 3 and Figure 4).

Peaks **1** and **4** were identified as delphinidin glycosides (Figure 2a and Appendix A). Peak **1** had a fragmentation pattern indicating the loss of galactose or glucose from *m*/*z* 465 ([M]^+^), while peak **4** (Rt = 7.79) showed a pattern consistent with arabinose dissociated from *m*/*z* 435 ([M]^+^). Delphinidin derivatives were eluted in the following order: peak **1** (Rt = 5.99) < peak **2** (delphinidin 3-*O*-glucoside) < peak **4** (Rt = 7.52). This elution order is consistent with the previously reported order of compounds with the same aglycone attached to sugars at the 3-OH position (galactose < glucose < arabinose) [32]. Consequently, peaks **1** and **4** were identified as delphinidin 3-*O*-galactoside and delphinidin 3-*O*-arabinoside, respectively.

Peaks **10** and **15** were found to be petunidin and malvidin glycoside, respectively (Appendix A). Based on the parent ion, aglycone ion, and differences in elution times corresponding to the linked sugars, they were further identified as petunidin 3-*O*-arabinoside (peak **10**) and malvidin 3-*O*-arabinoside (peak **15**).

Meanwhile, acetylated glycosides have been identified in blueberries with fragmented characteristics of acetyl-galactoside (42 + 162 Da) and acetyl-glucoside (42 + 162 Da), where deacetylation did not occur in positive mode [33]. A total of seven acetylated glycosides were identified. Among them, delphinidin 3-*O*-(6″-*O*-acetyl)glucoside (peak **16**), cyanidin 3-*O*-(6″-*O*-acetyl)glucoside (peak **18**), petunidin 3-*O*-(6″-*O*-acetyl)glucoside (peak **19**), and malvidin 3-*O*-(6″-*O*-acetyl)glucoside (peak **22**) have been previously reported in berries and wine [34]. Additionally, three acetylated glycosides were identified: petunidin 3-*O*-(6″-*O*-acetyl)galactoside (peak **17**), malvidin 3-*O*-(6″-*O*-acetyl)galactoside (peak **20**), and peonidin 3-*O*-(6″-*O*-acetyl)glucoside (peak **21**). Especially, the acetylated petunidin glycosides (peaks **17** and **19**) showed a parent ion at *m*/*z* 521 ([M]^+^), eluted similarly to peaks **6** (petunidin 3-*O*-galactoside) and **8** (petunidin 3-*O*-glucoside), but were detected later due to the acetyl group at the sugar’s 6″-OH position [31]. Thus, peaks **17** and **19** were identified as petunidin 3-*O*-(6″-*O*-acetyl)galactoside and petunidin 3-*O*-(6″-*O*-acetyl)glucoside, respectively (Figure 2).

Peaks **16** (*m*/*z* 507, 303), **18** (*m*/*z* 491, 287), **20** and **22** (*m*/*z* 535, 331), and **21** (*m*/*z* 505, 301) were assigned to delphinidin 3-*O*-(6″-*O*-acetyl)glucoside, cyanidin 3-*O*-(6″-*O*-acetyl)glucoside, malvidin 3-*O*-(6″-*O*-acetyl)galactoside and malvidin 3-*O*-(6″-*O*-acetyl)glucoside, and peonidin 3-*O*-(6″-*O*-acetyl)glucoside, respectively, based on their elution times, parent and aglycone ions, and characteristic MS spectra of acetylated glycosides (Figure 2).

### 3.2. Variation in Anthocyanin Contents Depending on Highbush Blueberry Cultivars

Table 5 presents the composition and content of 22 anthocyanins from 9 blueberry cultivars. Total anthocyanin content ranged from 581.1 to 1011.7 mg/100 g DW, which is similar to the reported ranges of 108.1–279.1 mg/100 g FW in China [35] and 65.5–267.84 mg/100 g FW in the United States and New Zealand [36]. In total contents, Legacy and New Hanover as mid-season cultivars had 725.4 and 1011.7 mg/100 g DW, respectively, and the early-season cultivars were observed in the following order: Patriot (910.4) > Spartan (855.5) > Draper (795.3) > Suziblue (771.1) > Reka (635.9) > Farthing (599.3) > Hannah’s Choice (581.1).

In the two mid-season cultivars (Legacy and New Hanover), malvidin glycosides accounted for 34% and 41% of total content, followed by delphinidin, petunidin, cyanidin, and peonidin derivatives. The seven early-season cultivars (Reka, Hannah’s Choice, Spartan, Draper, Suziblue, Farthing, and Patriot), accounting for malvidin 34–50%, delphinidin 24–32%, petunidin 20–26%, cyanidin 2–9%, and peonidin 1–5%, were similar to the mid-season cultivars. These results indicated that delphinidin, petunidin, and malvidin derivatives affected the total content regardless of cultivar and were also consistent with the aglycone proportions (delphinidin 27–40%, malvidin 22–33%, petunidin 19–26%, cyanidin 6–14%, and peonidin 1–5%) reported by Cho et al. [34]. The color of anthocyanins is influenced by the position and number of hydroxyl (-OH) or methoxyl (-OCH_3_) groups on the B-ring. Furthermore, Jung et al. [37] suggested that the biosynthetic pathway for malvidin is more dominant than that for cyanidin in highbush blueberries, which plays a crucial role in the berries’ blue color and antioxidant properties.

In other cultivars, except for ‘Patriot,’ delphinidin 3-*O*-galactoside, petunidin 3-*O*-galactoside, and malvidin 3-*O*-galactoside were identified as major components [11,12], and the ‘Patriot’ contained mainly glucose-bound anthocyanins in the present study, where galactose was the predominant glycoside in ‘Patriot’ blueberries provided from Italy and China [38,39]. The differences in composition and content of blueberry anthocyanins are considered to be generated by cultivars as well as various factors, such as cultivated conditions [40,41], temperature [19,42], UV radiation [43], and genetic factors [19,44].

The seven acetylated glycosides (peaks **16**–**22**) that accounted for 19.7% of total anthocyanins were highest in ‘Patriot’ (179.5), whereas these glycosides were not detected in Legacy, Draper, and Farthing (Table 5). Acetylated glycosides of ‘Patriot’ showed significant proportions of delphinidin and malvidin derivatives, which were consistent with data reported for Canadian lowbush and highbush blueberries [45]. Acetylated anthocyanins may have metabolic health benefits, including improved insulin sensitivity, reduced inflammation, and regulation of gut microbiota [46]. Malvidin 3-*O*-(6″-*O*-acetyl)glucoside has demonstrated free radical scavenging activity [47] and potential anti-inflammatory effects via selective COX-2 and *β*-glucosidase inhibition [16,48]. The synthesis of acylated anthocyanins can be catalyzed by acyltransferases, but the factors affecting the acetylation pathway are still unclear. Thus, further studies are required to elucidate the enzymatic activity and genetic regulation mechanism involved in the synthesis. In addition, the bioavailability, variations in metabolic processes, and biological activities of anthocyanins, including acetylated glycosides, should be evaluated in the future, and factors such as pH, light, and humidity need to be investigated to assess their stability and expand their applications.

### 3.3. Multivariate Statistical Analysis of Highbush Blueberry Cultivars

Multivariate statistical analysis was performed to visualize the differences in anthocyanins between highbush blueberry cultivars and to identify chemical markers contributing to these differences. The PCA score plot (R_2_X = 0.976, Q_2_ = 0.887) revealed three clusters (cluster 1, New Hanover; cluster 2, Draper, Legacy, Reka, Spartan, Suziblue, and Hannah’s Choice; cluster 3, Patriot). Patriot showed distinct differences from the other cultivars, primarily attributed to its high contents of acetyl glycosides. In the early season, Draper was found to share more similar compounds with the mid-season (Figure 5a). The PCA loading plot demonstrated variables corresponding to each cultivar (Appendix A). To further clarify the separation between two groups (early vs. mid) and identify differential markers, OPLS-DA was additionally conducted (R_2_X = 0.937, R_2_Y = 0.819, Q_2_ = 0.644). The OPLS analysis formed three clusters, with Draper positioned relatively close to the mid-season group, similar to the PCA results (Figure 5b). The OPLS loading plot highlighted variables with VIP > 1 in red, showing that compounds predominantly found in the early season were distributed in the second quadrant, and those in the mid-season were in the fourth quadrant, and these variables were identified as significant components distinguishing the cultivars (Appendix A). The accuracy and reliability of the 22 markers were validated through the AUCs and permutation plots for blueberries, indicating no overfitting in the model (Appendix A). The CV-ANOVA analysis yielded a *p*-value of 0.028 (*p* < 0.05), confirming that the established OPLS-DA model was statistically significant and reliable. The S-plot visualized the variables that distinguished the two groups; especially, delphinidin 3-*O*-galactoside (VIP = 1.94) was identified as a marker for mid-season, and malvidin 3-*O*-glucoside (VIP = 1.79) was identified as a marker for early season (Figure 5c and Appendix A). HCA was also performed to visualize the differences in content between early and mid-seasons for variables with VIP > 1, and similar to the OPLS-DA results, it distinguished two groups (early vs. mid). In conclusion, five markers (delphinidin 3-*O*-galactoside, delphinidin 3-*O*-arabinoside, petunidin 3-*O*-galactoside, malvidin 3-*O*-galactoside, and malvidin 3-*O*-arabinoside) of mid-seasons with Draper and three markers (delphinidin 3-*O*-glucoside, petunidin 3-*O*-glucoside, and malvidin 3-*O*-glucoside) of early seasons were confirmed (Figure 5d).

## 4. Conclusions

A total of 22 anthocyanins from 9 highbush blueberry cultivars included 4 delphinidin, 4 cyanidin, 5 petunidin, 4 peonidin, and 5 malvidin derivatives. ‘New Hanover’ (1011.7 mg/100 g DW) showed the highest total anthocyanin content among all cultivars, with malvidin and delphinidin glycosides accounting for over 70% of its total content. Especially, ‘Patriot’ (910.4), an early-season cultivar, recorded the highest proportion of acetylated anthocyanins (up to 19.7%), while non-acetylated anthocyanins were only detected in ‘Legacy,’ ‘Draper,’ and ‘Farthing.’ Multivariate analysis showed clear separation between early and mid-seasons with Draper. Among the eight markers, delphinidin 3-*O*-galactoside (VIP = 1.94) was identified as a marker for mid-season, and malvidin 3-*O*-glucoside (VIP = 1.79) was identified as a marker for early season. This study provided detailed chemical profiles on both major and trace components of acetylated anthocyanins from nine cultivars grown in Korea. Furthermore, these detailed findings will be suggested as fundamental data for the development of Korean superior cultivars and functional products, conditional variation evaluation, related fields, such as clinical and metabolomic research, as well as the further research of anthocyanin variations by harvest time, cultivated methods, and storage conditions.

## Figures and Tables

**Figure 1 foods-14-00188-f001:**
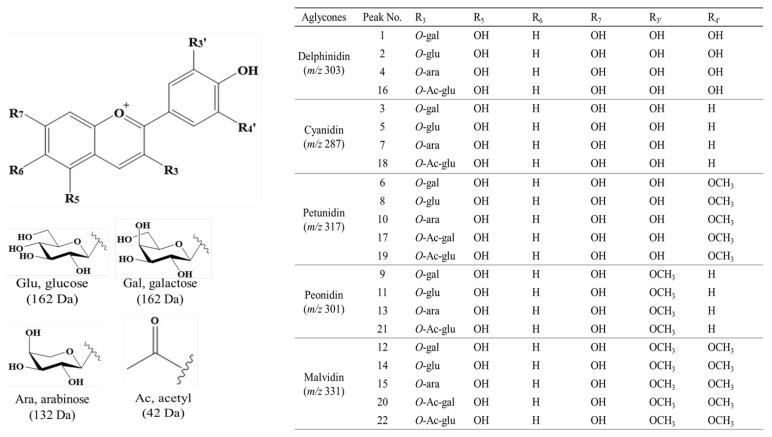
Chemical structures of 22 anthocyanin derivatives according to the positions of functional groups.

**Figure 2 foods-14-00188-f002:**
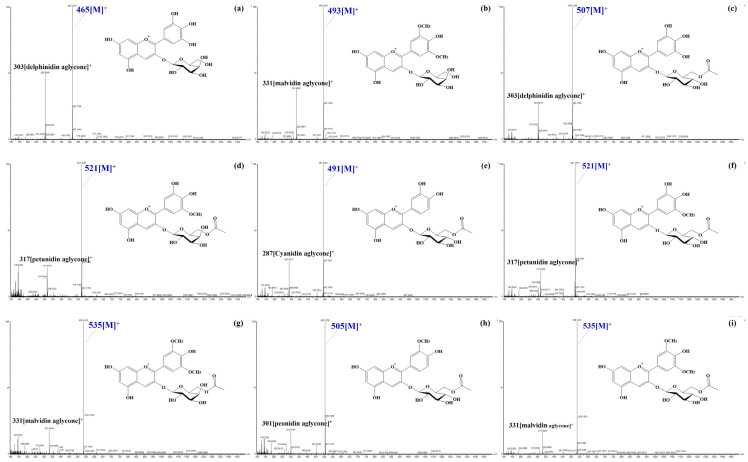
Fragmentation (*m*/*z*, [M]^+^) patterns of two major and seven acetylated anthocyanins identified in highbush blueberry: ((**a**), peak 1) Delphinidin 3-*O*-galactoside, ((**b**), peak 12) malvidin 3-*O*-galactoside, ((**c**), peak 16) delphinidin 3-*O*-(6″-*O*-acetyl)glucoside, ((**d**), peak 17) petunidin 3-*O*-(6″-*O*-acetyl)galactoside, ((**e**), peak 18) cyanidin 3-*O*-(6″-*O*-acetyl)glucoside, ((**f**), peak 19) petunidin 3-*O*-(6″-*O*-acetyl)glucoside, ((**g**), peak 20) malvidin 3-*O*-(6″-*O*-acetyl)galactoside, ((**h**), peak 21) peonidin 3-*O*-(6″-*O*-acetyl)glucoside, and ((**i**), peak 22) malvidin 3-*O*-(6″-*O*-acetyl)glucoside.

**Figure 3 foods-14-00188-f003:**
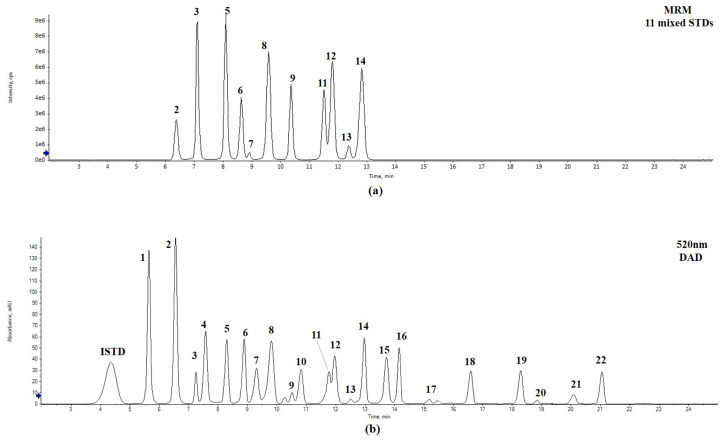
MRM chromatograms of 11 mixed standards (**a**) and UPLC-DAD chromatograms of 22 anthocyanins in Patriot (wavelength at 520 nm) (**b**). IS (internal standard): delphin 100 ppm.

**Figure 4 foods-14-00188-f004:**
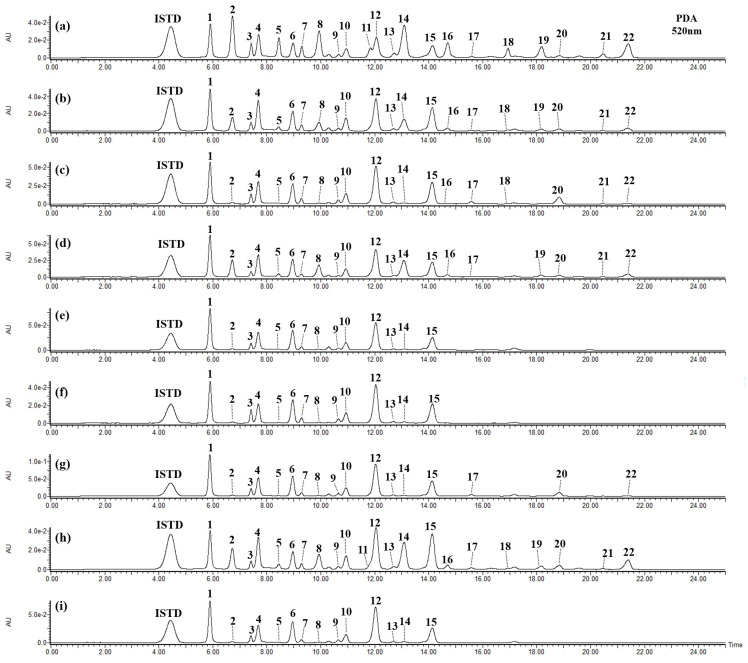
UPLC chromatograms of 22 anthocyanins (wavelength at 520 nm) from Patriot (**a**), Reka (**b**), Hannah’s Choice (**c**), Spartan (**d**), Legacy (**e**), Draper (**f**), New Hanover (**g**), Suziblue (**h**), and Farthing (**i**) highbush blueberry samples (*V. corymboums* L.) are presented according to peak number in Table 4. IS (internal standard): delphin 100 ppm.

**Figure 5 foods-14-00188-f005:**
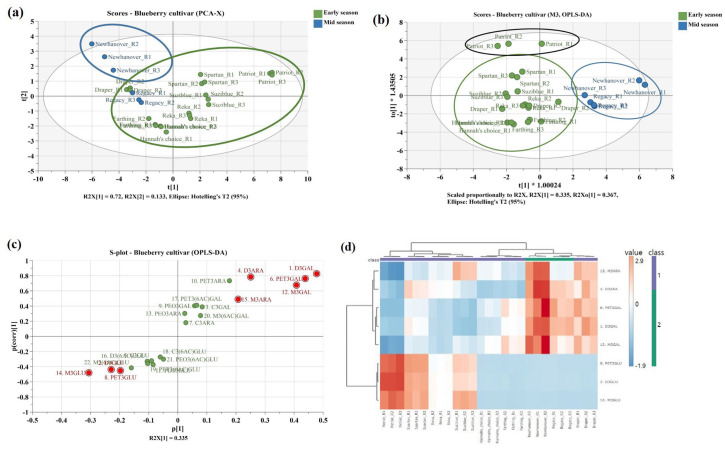
The multivariate statistical analysis of nine highbush blueberry cultivars. (**a**) PCA score plot, (**b**) OPLS-DA, (**c**) OPLS-DA loading S-plot, and (**d**) heat map of HCA.

**Table 1 foods-14-00188-t001:** List of nine highbush blueberry cultivars in Korea.

Cultivar	Harvest Season
Reka	Early
Hannah’s Choice	Early
Spartan	Early
Draper	Early
Patriot	Early
Legacy	Mid
Suziblue	Early
Farthing	Early
New Hanover	Mid

**Table 2 foods-14-00188-t002:** Optimized MRM conditions for quantitative analysis of blueberry anthocyanins.

No.	Compound Name	Formula	RT ^(1)^ (min)	Precursor ion (*m*/*z*)	Product ion (*m*/*z*)	DP (V) ^(2)^	CE (V) ^(3)^
2	Delphinidin 3-*O*-glucoside (mirtillin)	C_25_H_21_O_12_^+^	6.36	465	303	86	29
3	Cyanidin 3-*O*-galactoside (ideain)	C_21_H_21_O_11_^+^	7.09	449	287	86	33
5	Cyanidin 3-*O*-glucoside (asterin)	C_21_H_21_O_11_^+^	8.08	449	287	31	39
6	Petunidin 3-*O*-galactoside	C_22_H_23_O_12_^+^	8.62	479	317	86	35
7	Cyanidin 3-*O*-arabinoside	C_20_H_19_O_10_^+^	8.87	419	287	236	27
8	Petunidin 3-*O*-glucoside	C_22_H_23_O_12_^+^	9.57	479	317	86	31
9	Peonidin 3-*O*-galactoside	C_22_H_23_O_11_^+^	10.35	463	301	20	33
11	Peonidin 3-*O*-glucoside	C_22_H_23_O_11_^+^	11.50	463	301	20	33
12	Malvidin 3-*O*-galactoside (primulin)	C_22_H_25_O_12_^+^	11.79	493	331	11	35
13	Peonidin 3-*O*-arabinoside	C_21_H_21_O_10_^+^	12.36	433	301	104	33
14	Malvidin 3-*O*-glucoside (enin)	C_22_H_25_O_12_^+^	12.81	493	331	16	37

^(1)^ RT: retention time, ^(2)^ DP: de-clustering potential, and ^(3)^ CE: collision energy.

**Table 3 foods-14-00188-t003:** Linearity, regression equation, LOD and LOQ, and intra- and inter-day precisions of 11 anthocyanin standards.

Compound Name	Regression Equation	Correlation Coefficient ^(1)^ (R^2^)	LOD ^(2)^ (*µ*g/mL)	LOQ ^(3)^ (*µ*g/mL)	Precision RSD (%)	
					Intraday (*n* = 6)	Interday (*n* = 6)
Delphinidin 3-*O*-glucoside	*Y* = 1779297.2*X* + 9835.2387	0.9999	0.0025	0.0076	1.83	6.54
Cyanidin 3-*O*-galactoside	*Y* = 2178374.9*X* + 25655.6942	1.0000	0.0026	0.0080	0.50	5.63
Cyanidin 3-*O*-glucoside	*Y* = 3597678.5*X* + 27299.5857	1.0000	0.0030	0.0091	0.03	5.27
Petunidin 3-*O*-galactoside	*Y* = 1184176.8*X* + 8718.2247	0.9998	0.0011	0.0033	5.90	4.34
Cyanidin 3-*O*-arabinoside	*Y* = 532110.7*X* + 5725.4070	0.9999	0.0039	0.0117	0.50	1.96
Petunidin 3-*O*-glucoside	*Y* = 2320782.4*X* + 22315.0959	0.9999	0.0017	0.0053	0.03	3.42
Peonidin 3-*O*-galactoside	*Y* = 1537207.3*X* + 46.9756	0.9999	0.0035	0.0106	5.43	3.21
Peonidin 3-*O*-glucoside	*Y* = 1639444.7*X* + 6829.8984	0.9999	0.0035	0.0105	0.49	4.57
Malvidin 3-*O*-galactoside	*Y* = 3097011.3*X* + 49762.1926	0.9998	0.0042	0.0128	0.03	3.31
Peonidin 3-*O*-arabinoside	*Y* = 817531.0*X* + 15169.9675	0.9998	0.0028	0.0085	6.69	3.84
Malvidin 3-*O*-glucoside	*Y* = 3094396.6*X* + 62198.3776	0.9998	0.0017	0.0051	0.49	3.95

^(1)^ Correlation coefficient (R^2^) of calibration curve (range: 0.01–5 µg/mL), *Y* = peak area, *X* = concentration (ppm), ^(2)^ LOD: limit of detection, and ^(3)^ LOQ: limit of quantitation.

**Table 4 foods-14-00188-t004:** Twenty-two anthocyanins identified from highbush blueberry cultivars.

Peak No.	RT (min)	Compound Name	Formula	Experimental ion (*m*/*z*, [M]^+^)	Error (ppm) ^(1)^	Productions (*m*/*z*)	Species ^(2)^
1	5.99	Delphinidin 3-*O*-galactoside	C_21_H_21_O_12_^+^	465.1022	−1.2	465, 303	a, b, c, d, e, f, g, h, i
2	6.82	Delphinidin 3-*O*-glucoside	C_25_H_21_O_12_^+^	465.1021	−1.4	465, 303	a, b, c, d, e, f, g, h, i
3	7.52	Cyanidin 3-*O*-galactoside	C_21_H_21_O_11_^+^	449.1080	0.4	449, 287	a, b, c, d, e, f, g, h, i
4	7.79	Delphinidin 3-*O*-arabinoside	C_20_H_19_O_11_^+^	435.0922	0.0	435, 303	a, b, c, d, e, f, g, h, i
5	8.56	Cyanidin 3-*O*-glucoside	C_21_H_21_O_11_^+^	449.1080	0.4	449, 287	a, b, c, d, e, f, g, h, i
6	9.09	Petunidin 3-*O*-galactoside	C_22_H_23_O_12_^+^	479.1184	0.0	479, 317	a, b, c, d, e, f, g, h, i
7	9.41	Cyanidin 3-*O*-arabinoside	C_20_H_19_O_10_^+^	419.0975	0.5	419, 287	a, b, c, d, e, f, g, h, i
8	10.06	Petunidin 3-*O*-glucoside	C_22_H_23_O_12_^+^	479.1185	0.2	479, 317	a, b, c, d, e, f, g, h, i
9	10.79	Peonidin 3-*O*-galactoside	C_22_H_23_O_11_^+^	463.1270	0.5	463, 301	a, b, c, d, e, f, g, h, i
10	11.08	Petunidin 3-*O*-arabinoside	C_21_H_21_O_11_^+^	449.1078	−0.1	449, 317	a, b, c, d, e, f, g, h, i
11	11.98	Peonidin 3-*O*-glucoside	C_22_H_23_O_11_^+^	463.1234	−0.2	463, 301	e, g
12	12.20	Malvidin 3-*O*-galactoside	C_22_H_25_O_12_^+^	493.1337	−0.7	493, 331	a, b, c, d, e, f, g, h, i
13	12.85	Peonidin 3-*O*-arabinoside	C_21_H_21_O_10_^+^	433.1133	0.9	433, 301	a, b, c, d, e, f, g, h, i
14	13.25	Malvidin 3-*O*-glucoside	C_22_H_25_O_12_^+^	493.1342	0.3	493, 331	a, b, c, d, e, f, g, h, i
15	14.30	Malvidin 3-*O*-arabinoside	C_22_H_23_O_11_^+^	463.1235	0.0	463, 331	a, b, c, d, e, f, g, h, i
16	14.88	Delphinidin 3-*O*-(6″-*O*-acetyl)glucoside	C_23_H_23_O_13_^+^	507.1135	0.4	507, 303	a, b, c, e, g
17	15.78	Petunidin 3-*O*-(6″-*O*-acetyl)galactoside	C_24_H_25_O_13_^+^	521.1292	0.4	521, 317	a, b, c, e, g, i
18	17.13	Cyanidin 3-*O*-(6″-*O*-acetyl)glucoside	C_23_H_23_O_12_^+^	491.1187	0.6	491, 287	a, b, c, e, g
19	18.39	Petunidin 3-*O*-(6″-*O*-acetyl)glucoside	C_24_H_25_O_13_^+^	521.1292	0.4	521, 317	a, b, c, e, g
20	19.06	Malvidin 3-*O*-(6″-*O*-acetyl)galactoside	C_25_H_27_O_13_^+^	535.1449	0.5	535, 331	a, b, c, e, g, i
21	20.69	Peonidin 3-*O*-(6″-*O*-acetyl)glucoside	C_24_H_25_O_12_^+^	505.1343	0.4	505, 301	a, b, c, e, g
22	21.62	Malvidin 3-*O*-(6″-*O*-acetyl)glucoside	C_25_H_27_O_13_^+^	535.1448	0.3	535, 331	a, b, c, e, g, i

^(1)^ Error (ppm) indicates the mass accuracy of QToF data and was calculated as: [(calculated ion − observed ion)/(calculated ion)] × 10^6^, based on *m*/*z* [M+H]^+^. ^(2)^ a, Reka; b, Hannah’s Choice; c, Spartan; d, Draper; e, Patriot; f, Legacy; g, Suziblue; h, Farthing; i, New Hanover.

**Table 5 foods-14-00188-t005:** Contents of individual anthocyanin glycosides in different highbush blueberry (*Vaccinum corymbosum L.*) cultivars.

Peak No. ^(1)^	Anthocyanin Content (mg/100 g DW)								
	Reka	Hannah’s Choice	Spartan	Suziblue	Farthing	Patriot	Draper	Legacy	New Hanover
	Early Season							Mid-Season	
1	86.7 ± 1.4 ^d^	94.7 ± 1.8 ^d^	123.4 ± 7.7 ^c^	71.2 ± 2.1 ^e^	119.7 ± 2.6 ^c^	72.8 ± 2.5 ^e^	164.4 ± 19.4 ^b^	165.1 ± 3.1 ^b^	200.4 ± 16.8 ^a^
2 *	33.2 ± 1.9 ^d^	4.1 ± 0.2 ^e^	74.2 ± 2.2 ^b^	49.0 ± 2.0 ^c^	3.3 ± 1.3 ^e^	110.0 ± 0.9 ^a^	4.0 ± 0.6 ^e^	4.8 ± 0.7 ^e^	4.8 ± 0.4 ^e^
3 *	10.7 ± 0.7 ^fg^	14.8 ± 0.5 ^de^	8.3 ± 0.1 ^e^	9.0 ± 0.5 ^ge^	12.2 ± 1.4 ^ef^	17.0 ± 1.6 ^c^	27.7 ± 3.0 ^a^	16.2 ± 2.2 ^cd^	22.8 ± 2.5 ^b^
4	65.7 ± 0.9 ^cd^	51.8 ± 0.9 ^e^	71.1 ± 2.9 ^bc^	59.5 ± 2.7 ^d^	52.0 ± 1.5 ^e^	49.4 ± 2.6 ^e^	74.3 ± 6.7 ^b^	74.4 ± 1.5 ^b^	91.7 ± 7.4 ^a^
5 *	5.4 ± 0.7 ^b^	0.7 ± 0.2 ^c^	6.9 ± 0.2 ^b^	6.2 ± 0.5 ^b^	0.4 ± 0.1 ^c^	33.7 ± 2.3 ^a^	0.7 ± 0.2 ^c^	0.6 ± 0.2 ^c^	0.6 ± 0.1 ^c^
6 *	84.4 ± 3.7 ^de^	102.7 ± 5.9 ^d^	108.0 ± 1.5 ^d^	73.8 ± 6.0 ^ef^	127.8 ± 10.3 ^c^	61.9 ± 0.3 ^f^	159.4 ± 13.4 ^b^	157.3 ± 14.6 ^b^	193.6 ± 18.3 ^a^
7 *	7.5 ± 0.8 ^c^	7.3 ± 1.2 ^c^	3.6 ± 0.3 ^f^	6.0 ± 0.5 ^de^	5.3 ± 0.4 ^e^	13.2 ± 0.9 ^a^	11.0 ± 0.1 ^b^	7.0 ± 0.6 ^cd^	10.6 ± 0.8 ^b^
8 *	22.5 ± 2.1 ^d^	2.8 ± 0.0 ^e^	56.0 ± 1.0 ^b^	38.6 ± 2.6 ^c^	2.7 ± 0.1 ^e^	72.1 ± 3.9 ^a^	3.4 ± 0.1 ^e^	3.8 ± 0.3 ^e^	3.8 ± 0.3 ^e^
9 *	5.0 ± 1.0 ^ef^	8.6 ± 2.1 ^b^	3.4 ± 0.1 ^g^	4.5 ± 0.2 ^f^	7.5 ± 1.1 ^bc^	6.1 ± 0.3 ^de^	11.2 ± 0.8 ^a^	6.7 ± 0.6 ^cd^	11.2 ± 1.0 ^a^
10	35.4 ± 0.5 ^d^	27.1 ± 0.5 ^e^	32.8 ± 1.8 ^d^	32.5 ± 1.8 ^d^	27.8 ± 1.0 ^e^	22.7 ± 1.3 ^f^	42.6 ± 1.4 ^b^	39.4 ± 0.8 ^c^	47.5 ± 3.5 ^a^
11 *	4.4 ± 0.7 ^d^	1.0 ± 0.3 ^e^	5.8 ± 0.6 ^c^	7.7 ± 0.5 ^b^	0.7 ± 0.1 ^e^	23.1 ± 1.4 ^a^	0.9 ± 0.2 ^e^	0.8 ± 0.1 ^e^	0.9 ± 0.1 ^e^
12 *	103.5 ± 3.2 ^d^	133.8 ± 11.7 ^c^	121.8 ± 3.1 ^cd^	120.9 ± 7.3 ^cd^	157.9 ± 14.0 ^b^	59.9 ± 3.7 ^e^	177.9 ± 15.9 ^b^	157.7 ± 13.1 ^b^	238.1 ± 22.7 ^a^
13 *	0.8 ± 0.5 ^e^	2.2 ± 0.2 ^c^	ND ^f^	0.7 ± 0.1 ^e^	1.5 ± 0.1 ^d^	1.9 ± 0.1 ^c^	2.2 ± 0.2 ^b^	1.2 ± 0.1 ^d^	2.7 ± 1.1 ^a^
14 *	49.7 ± 2.6 ^c^	3.9 ± 0.3 ^d^	109.4 ± 7.9 ^b^	110.5 ± 6.6 ^b^	4.0 ± 0.4 ^d^	140.9 ± 9.0 ^a^	4.5 ± 0.3 ^d^	4.7 ± 0.4 ^d^	5.8 ± 0.4 ^d^
15	85.1 ± 1.5 ^c^	87.1 ± 1.7 ^c^	78.7 ± 3.6 ^d^	109.7 ± 5.8 ^b^	76.4 ± 2.0 ^d^	46.2 ± 3.4 ^e^	111.1 ± 4.7 ^b^	85.6 ± 0.5 ^c^	133.2 ± 7.5 ^a^
16	6.9 ± 0.1 ^c^	0.9 ± 0.3 ^d^	9.0 ± 0.5 ^b^	9.0 ± 0.6 ^b^	ND ^d^	43.9 ± 2.8 ^a^	ND ^d^	ND ^d^	ND ^d^
17	1.4 ± 0.0 ^e^	6.5 ± 0.4 ^b^	1.7 ± 0.3 ^de^	3.2 ± 0.1 ^c^	ND ^f^	2.1 ± 0.2 ^d^	ND ^f^	ND ^f^	10.5 ± 0.9 ^a^
18	2.3 ± 0.5 ^b^	0.9 ± 0.1 ^de^	1.3 ± 0.1 ^cd^	2.0 ± 0.2 ^bc^	ND ^e^	21.0 ± 1.1 ^a^	ND ^e^	ND ^e^	ND ^e^
19	5.8 ± 0.4 ^c^	ND ^d^	9.8 ± 0.4 ^b^	9.3 ± 0.6 ^b^	ND ^d^	36.0 ± 1.5 ^a^	ND ^d^	ND ^d^	ND ^d^
20	6.7 ± 0.2 ^e^	26.9 ± 0.5 ^b^	8.9 ± 0.3 ^d^	11.8 ± 0.8 ^c^	ND ^f^	6.2 ± 0.3 ^e^	ND ^f^	ND ^f^	31.6 ± 2.0 ^a^
21	1.3 ± 0.0 ^c^	0.7 ± 0.0 ^d^	0.9 ± 0.1 ^cd^	2.1 ± 0.2 ^b^	ND ^e^	11.2 ± 0.7 ^a^	ND ^e^	ND ^e^	ND ^e^
22	11.5 ± 0.0 ^d^	2.7 ± 0.1 ^e^	20.7 ± 0.5 ^c^	34.0 ± 1.8 ^b^	ND ^f^	59.2 ± 3.9 ^a^	ND ^f^	ND ^f^	1.9 ± 0.1 ^e^
Total anthocyanins	635.9 ± 18.9 ^d^	581.1 ± 23.8 ^de^	855.5 ± 27.3 ^bc^	771.1 ± 32.7 ^c^	599.3 ± 12.5 ^e^	910.4 ± 36.8 ^ab^	795.3 ± 31.5 ^c^	725.4 ± 36.8 ^ab^	1011.7 ± 66.4 ^a^

^(1)^ 1, delphinidin 3-*O*-galactoside; 2, delphinidin 3-*O*-glucoside; 3, cyanidin 3-*O*-galactoside; 4, delphinidin 3-*O*-arabinoside; 5, cyanidin 3-*O*-glucoside; 6, petunidin 3-*O*-galactoside; 7, cyanidin 3-*O*-arabinoside; 8, petunidin 3-*O*-glucoside; 9, peonidin 3-*O*-galactoside; 10, petunidin 3-*O*-arabinoside; 11, peonidin 3-*O*-glucoside; 12, malvidin 3-*O*-galactoside; 13, peonidin 3-*O*-arabinoside; 14, malvidin 3-*O*-glucoside; 15, malvidin 3-*O*-arabinoside; 16, delphinidin 3-*O*-(6″-*O*-acetyl)glucoside; 17, petunidin 3-*O*-(6″-*O*-acetyl)galactoside; 18, cyanidin 3-*O*-(6″-*O*-acetyl)glucoside; 19, petunidin 3-*O*-(6″-*O*-acetyl)glucoside; 20, malvidin 3-*O*-(6″-*O*-acetyl)galactoside; 21, peonidin 3-*O*-(6″-*O*-acetyl)glucoside; 22, malvidin 3-*O*-(6″-*O*-acetyl)glucoside. Each value was calculated as mean ± SD (*n* = 3) using an internal standard (delphin) and * external standards. ^a–f^ Different superscript letters next to mean values (*n* = 3) indicate significant differences (*p* < 0.05) according to Duncan’s multiple range test. ND means “not detected”.

## Data Availability

The original contributions presented in this study are included in the article/Appendix A. Further inquiries can be directed to the corresponding authors.

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
