# Peer review of "Characterization of Anthocyanins Including Acetylated Glycosides from Highbush Blueberry (Vaccinium corymbosum L.) Cultivated in Korea Based on UPLC-DAD-QToF/MS and UPLC-Qtrap-MS/MS"

_foods, 2025, doi:10.3390/foods14020188_

Round 1
Reviewer 1 Report
Comments and Suggestions for Authors
In the manuscript submitted for review, the Authors characterized 9 highbush blueberry varieties in terms of anthocyanin content. They analyzed 22 identified derivatives using liquid chromatography coupled with a mass detector.
Although I consider the manuscript to be correct in technical terms, in my opinion it does not demonstrate much originality in terms of the presented results.
The Authors devoted a lot of attention to the issue of describing the anthocyanin profile, where this knowledge is very well confirmed by other authors in numerous publications. The Authors devoted too much attention to this part, where in fact the anthocyanin profile is very simple and obvious to interpret.
My suggestions for the Authors.
1. Justification and broader explanation of the innovativeness and originality of the undertaken research
2. Broader discussion of the obtained results in terms of comparing them with other varieties and with varieties grown in other countries and climate zones.
3. The manuscript is very poor in statistical analyses, it suggests using more precise analysis methods such as PCA and hierarchical analyses, which will explain in more detail the differences and similarities in the profile of the tested varieties
4. Why the Authors focused only on anthocyanins, this is not explained in the aim of the work. And what about other classes of polyphenolic compounds
Author Response
Response to reviewer's comments
Thanks very much for the reviewer's comments and kind advice on our manuscript. All of these comments have helped us so much to improve the manuscript. We have studied these comments carefully, and our responses to all the comments are given below in a way of point by point. Besides, some changes have been made throughout the whole manuscript, and all the modifications, corrections and changes are highlighted by red color in the revised manuscript.
C1. Justification and broader explanation of the innovativeness and originality of the undertaken
(Author’s response)
Thank you for your comment. This study reported the composition and content of anthocyanins in nine cultivars from Korea and provided additional information on acetylated anthocyanins, complementing previous research on blueberry anthocyanins. In addition, multivariate statistical analysis was performed to visualize the differences between blueberry cultivars and identified discriminant markers critical for cultivar differentiation Therefore, the author revised and added the corresponding text in the abstract (lines 24-27) introduction (lines 79-80), materials and methods (line 162-175), results and discussion (line 300-329), Conclusion (lines 340-342) and additionally presented Figure 5, Figure S2, Figure S3 and Table S1.
(After) In addition, multivariate analysis visualized differences between cultivars and identified discriminant markers critical for cultivar differentiation. (lines 79-80)
C2. Broader discussion of the obtained results in terms of comparing them with other varieties and with varieties grown in other countries and climate zones.
(Author’s response)
Thank you for your comment. We agree with your opinion and added an explanation to provide a broader discussion in results and discussion Lines 249-251, Lines 281-283
(After)
Total anthocyanin content ranged from 581.1 to 1011.7 mg/100 g DW, which is similar to the reported ranges of 108.1–279.1 mg/100 g FW in China [35] and 65.5–267.84 mg/100 g FW in the United States and New Zealand. [36]. (Lines 249-251)
The differences in composition and content of blueberry anthocyanins are considered to be generated by cultivars as well as various factors, such as cultivated conditions [40,41], temperature [19,42], UV radiation [43], and genetic factors [19,44]. (Lines 281-283)
[36] Rossi, G.; Woods, F.M.; Leisner, C.P. Quantification of total phenolic, anthocyanin, and flavonoid content in a diverse panel of blueberry cultivars and ecotypes. HortScience 2022, 57, 901-909
[19] Spinardi, A.; Cola, G.; Gardana, C.S.; Mignani, I. Variation of anthocyanin content and profile throughout fruit development and ripening of highbush blueberry cultivars grown at two different altitudes. Front. Plant Sci. 2019, 10, 1045.
C3. The manuscript is very poor in statistical analyses, it suggests using more precise analysis methods such as PCA and hierarchical analyses, which will explain in more detail the differences and similarities in the profile of the tested varieties3.
(Author’s response)
Thank you for your comment. We agree with the reviewer's comment that the statistical analysis was insufficient. Therefore, we have added relevant information as suggested abstract (ines 24-27), introduction (lines 79-80), materials and methods (ines 162-175), results and discussion (lines 300-329), conclusions (lines 340-343) and additionally presented Figure 5, Figure S2, Figure S3 and Table S1.
(After)
Abstract: Multivariate analysis showed a distinct separation between early- and mid-seasons with Draper. Especially, delphinidin 3-O-galactoside (VIP = 1.94) was identified as a marker for mid-season, malvidin 3-O-glucoside (VIP = 1.79) was identified as a marker for early-season. (lines 24-27)
Method: 2.5. Muliple anlysis for anthocyanins
A multivariate analysis was conducted using SIMCA (version 16, Satorius Stedim Data Analysis AB, Sweden). Principal component analysis (PCA) and orthogonal partial least squares discriminant analysis (OPLS-DA) were applied for statistical comparison and marker identification of nine blueberry cultivars. The scaling method was performed using Pareto (par). Chemical markers that distinguish the two species were obtained by combining S- and variable importance in projection (VIP) plots. Chemical markers were selected with VIP > 1 and p-value < 0.05 to indicate their influence on classification. To evaluate the reliability and predictability of the OPLS-DA model, 7-fold cross validation (CV) and 200 random permutation tests, and receiver operating characteristic (ROC) curve models were performed. Additionally, hierarchical clustering analysis (HCA) with a heat map was constructed using Euclidean distance method using MetaboAnalyst online analysis software (http://www.metaboanalyst.ca) for clustering and visualization of the chemical markers between early and mid-seasons. (lines 162-175)
Results and Dis:cussion: 3.3. Multivariate statistical analysis of highbush blueberry cultivars
Multivariate statistical analysis was performed to visualize the differences of anthocyanin between highbush blueberry cultivars and to identify chemical markers contributing to these differences. The PCA score plot (R2X = 0.976, Q2 = 0.887) revealed three clusters (cluster 1, New hanover; cluster 2, Draper, Legacy, Reka, Spartan, Suziblue and Hannah's choice; cluster 3, Patriot). Patriot showed distinct differences from other cultivars, primarily attributed to its high contents of acetyl glycosides. As early-season, Draper was found to share more similar compounds with mid-season (Figure 5a). The PCA loading plot demonstrated variables corresponding to each cultivar (Figure S1a). To further clarify the separation between two groups (early vs. mid) and identify differential markers, OPLS-DA analysis was additionally conducted (R2X = 0.937, R2Y = 0.819, Q2 = 0.644). The OPLS analysis formed three clusters, with Draper positioned relatively close to the mid-season group, similar to the PCA results (Figure 5b). The OPLS loading plot highlighted variables with VIP > 1 in red, showing that compounds predominantly found in early-season were distributed in the second quadrant, and those in mid-season were in the fourth quadrant, and these variables were identified as significant components distinguishing the cultivars (Figure S1b). The accuracy and reliability of the 22 markers were validated through the AUCs and permutation plots for blueberries, indicating no overfitting in the model (Figure S2). The CV-ANOVA analysis yielded a p-value of 0.028 (p < 0.05), confirming that the established OPLS-DA model is statistically significant and reliable. The S-plot visualized the variables that distinguish the two groups, especially, delphinidin 3-O-galactoside (VIP = 1.94) was identified as a marker for mid-season, while malvidin 3-O-glucoside (VIP = 1.79) was identified as a marker for early-season (Figure 5c, Table S1). HCA was also performed to visualize the differences in content between early and mid-seasons for variables with VIP > 1, and similar to the OPLS-DA results, it distinguished two groups (early vs. mid). In conclusion, five markers (delphinidin 3-O-galactoside, delphinidin 3-O-arabinoside, petunidin 3-O-galactoside, malvidin 3-O-galactoside and malvidin 3-O-arabinoside) of mid-seasons with Draper and three markers (delphinidin 3-O-glucoside, petunidin 3-O-glucoside and malvidin 3-O-glucoside) of early seasons were confirmed (Figure 5d). (lines 300-329)
Conclusion: Multivariate analysis showed clear separation between early- and mid-seasons with Draper. Among the 8 markers, delphinidin 3-O-galactoside (VIP = 1.94) was identified as a marker for mid-season, malvidin 3-O-glucoside (VIP = 1.79) was identified as a marker for early-season. (lines 340-343)
Figure 5. The multivariate statistical analysis of 9 highbush blueberry cultivars. (a) PCA score plot, (b) OPLS-DA score pot, (c) OPLS-DA S-plot, (d), and heat map of HCA.
Figure S2. Validation of the OPLS-DA model for highbush blueberry cultivars (a) ROC curve, (b) permutation test for early-season and (c) permutation test for mid-season.
Figure S3. Multivariate analyses in 9 highbush blueberry cultivars based on MS data. (a) PCA loading plot and (b) OPLS-DA loading plot.
Table S1. VIP values and p-values of markers identified between early and mid-seasons by OPLS-DA and t-test
Compound |
1)VIP-value |
2)p-value |
Delphinidin 3-O-galactoside |
1.94 |
1.98E-05 |
Delphinidin 3-O-glucoside |
1.84 |
4.06E-02 |
Delphinidin 3-O-arabinoside |
1.79 |
5.61E-04 |
Petunidin 3-O-galactoside |
1.61 |
5.67E-05 |
Petunidin 3-O-glucoside |
1.29 |
1.86E-03 |
Malvidin 3-O-galactoside |
1.08 |
4.39E-05 |
Malvidin 3-O-glucoside |
1.08 |
6.05E-04 |
Malvidin 3-O- arabinoside |
1.02 |
3.64E-02 |
1) VIP was obtained from OPLS-DA with a threshold of 1.
2) p-value was calculated with t-test
C4. Why the Authors focused only on anthocyanins, this is not explained in the aim of the work. And what about other classes of polyphenolic compounds
(Author’s response)
Thank you for your feedback. Anthocyanins are the most abundant and characteristic polyphenols in blueberries, accounting for 35-74% of total phenols. They play a key role in determining the primary pigments of blueberries, blue and purple, and have been extensively studied for their various physiological activities, including antioxidant, anti-inflammatory, and cardiovascular disease prevention effects. However, hydroxycinnamic acid derivatives, flavan-3-ols, and flavonols have been reported in relatively small amounts compared to anthocyanins. Therefore, we focused on anthocyanin. According to your suggestion, we will conduct research on the characteristics of the previously mentioned components (hydroxycinnamic acid derivatives, flavan-3-ols, and flavonols) with anthocyanins in the future. The text has been added to the Introduction (lines 46-48) with appropriate references.
(After) ~ Anthocyanins are the most abundant and characteristic polyphenols in blueberries, accounting for 35-74% of the total phenolics in blueberries [10].
[10] Cosme, F.; Gonçalves, B.; Bacelar, E.A.; Inês, A.; Jordão, A.M.; Vilela, A. Genotype, environment and management practices on red/dark-colored fruits phenolic composition and its impact on sensory attributes and potential health benefits. In Phenolic Compounds-Natural Sources, Importance and Applications; IntechOpen: 2017.

Reviewer 2 Report
Comments and Suggestions for Authors
This study offers a comprehensive analysis of anthocyanin profiles placing a particular emphasis on the less-explored acetylated glycosides. Utilizing advanced analytical techniques, including UPLC-DAD-QToF/MS and UPLC-QTrap-MS/MS, the research provides precise identification and quantification of these compounds. The findings have significant effects, contributing to the development of functional blueberry-based products, and broader applications in the functional food industry.
I have some minor comments:
Include the significance of acetylated glycosides and emphasize their antioxidant potential and application.
Rephrase the sentence, "These detailed anthocyanin profiles will be supported as fundamental data..." for clarity.
Focus on Korean cultivars is significant compared to previous studies in other regions.
Ensure the use of consistent units (mg/100 g DW).
A table can be constructed to present the health-related activities of identified anthocyanins.
Strengthen the link between findings and potential industrial applications.
Comment on future studies on environmental or storage conditions affecting acetylated anthocyanin stability.
Author Response
Response to reviewer's comments
Thanks very much for the reviewer's comments and kind advice on our manuscript. All of these comments have helped us so much to improve the manuscript. We have studied these comments carefully, and our responses to all the comments are given below in a way of point by point. Besides, some changes have been made throughout the whole manuscript, and all the modifications, corrections and changes are highlighted by red color in the revised manuscript.
C1. Include the significance of acetylated glycosides and emphasize their antioxidant potential and application.
(Author’s response)
Thank you for your comment. We agree with your opinion and the sentence was modified as follow (Lines 56-59)
(After) These compounds are reported to contribute to the intensity, stability of pigments in foods [12] and exhibit antioxidant and β-glucosidase effects [15,16], suggesting applications in functional foods and pharmaceuticals [17].
[16] Wu, Q.; Zhang, Y.; Tang, H.; Chen, Y.; Xie, B.; Wang, C.; Sun, Z. Separation and Identification of Anthocyanins Extracted from Blueberry Wine Lees and Pigment Binding Properties toward beta-Glucosidase. J. Agric. Food Chem. 2017, 65, 216-223.
[17] Ananga, A.; Georgiev, V.; Ochieng, J.; Phills, B.; Tsolova, V. Production of anthocyanins in grape cell cultures: a potential source of raw material for pharmaceutical, food, and cosmetic industries. The Mediterranean genetic code-grapevine and olive 2013, 1, 247-287.
C2. Rephrase the sentence, "These detailed anthocyanin profiles will be supported as fundamental data..." for clarity.
(Author’s response)
Thank you for your comment. We agree with your opinion and the sentence was modified as follow abstract (Lines 27-29)
(After) Abstract: These comprehensive anthocyanin profiles of Korean blueberries will serve as fundamental data for breeding superior cultivars, evaluating and developing related products as well as clinical and metabolomic research. (Lines 27-29)
C3. Focus on Korean cultivars is significant compared to previous studies in other regions.
(Author’s response)
Thank you for your comment. We agree with your opinion and added an explanation to provide a broader discussion in results and discussion Lines 249-251, Lines 281-283
(After)
Total anthocyanin content ranged from 581.1 to 1011.7 mg/100 g DW, which is similar to the reported ranges of 108.1–279.1 mg/100 g FW in China [35] and 65.5–267.84 mg/100 g FW in the United States and New Zealand. [36]. (Lines 249-251)
The differences in composition and content of blueberry anthocyanins are considered to be generated by cultivars as well as various factors, such as cultivated conditions [40,41], temperature [19,42], UV radiation [43], and genetic factors [19,44]. (Lines 281-283)
[36] Rossi, G.; Woods, F.M.; Leisner, C.P. Quantification of total phenolic, anthocyanin, and flavonoid content in a diverse panel of blueberry cultivars and ecotypes. HortScience 2022, 57, 901-909
[19] Spinardi, A.; Cola, G.; Gardana, C.S.; Mignani, I. Variation of anthocyanin content and profile throughout fruit development and ripening of highbush blueberry cultivars grown at two different altitudes. Front. Plant Sci. 2019, 10, 1045.
C4. Ensure the use of consistent units (mg/100 g DW).
(Author’s response)
Thank you for your comment. The units (mg/100 g DW) has been modified Line 138, Line 249, Line 252, Line 336 and Table 5.
C5. A table can be constructed to present the health-related activities of identified anthocyanins.
(Author’s response)
Thank you for your comments. Based on your suggestions, a table summarizing the health-related activities of the identified anthocyanins has been provided. Additionally, this information has been included as Supplementary Table S2. (Line 54)
(After)
Table S2. Biological activities of various anthocyanins
Anthocyanin. |
Model |
Does or concentrations |
Antioxidant |
Mechanism |
Reference |
Delphinidin 3-O-glucoside |
HCT-116 colorectal cancer cell line |
100-600 mg/mL |
Anticancer |
induce PD-1 expression inhibition and PD-L1 binding reduction, leading to decreased cancer cell proliferation |
[49] |
Molecular Docking, TNF-α Signaling Assay |
- |
Anti-inflammation |
Inhibition TNF-α receptor and TNF-α signaling by direct interaction Docking energy: -9.21 kcal/mol |
[50] |
|
Cyanidin 3-O-glucoside |
Co-culture with PBMC and HCT 116 cells |
100 mM |
Anticancer |
inhibition PD-1/PD-L1 expression, reduction VEGF levels and blocking immune evasion and cancer cell survival |
[49] |
Malvidin 3-O-galactoside Malvidin 3-O-glucoside |
human umbilical vein endothelial (HUVECs) |
10 mM |
Cardioprotective |
Decreased MCP-1, ICAM-1, and VCAM-1 expression Inhibition of IκBα degradation and p65 nuclear translocation |
[51] |
Cyanidin 3-O-rhamnoside |
human liver microsomes (pooled) |
0-100 mM |
Protective hepatocyte |
Inhibition CYP3A4 activity (IC50 = 44 μM), Regulation CYP2C9, CYP2A6, and CYP2B6 |
[52] |
Delphinidin 3-O-rutinoside |
human liver microsomes (pooled) |
0-100 mM |
Protective hepatocyte |
Inhibition CYP3A4 activity (down to 35% at 100 μM, IC50 = 67 μM) |
[52] |
Malvidin 3-O-(6''-O-acetyl)glucoside |
β-glucosidase inhibition assay |
0-90 mM |
Anti-inflammation |
inhibition of β-glucosidase activity and enhanced structural changes in the enzyme |
[16] |
Peonidin 3-O-glucoside |
Molecular Docking, TNF-α Signaling assay |
- |
Anti-inflammation |
Inhibition the interaction between TNF-α receptor and TNF-α protein, stabilizing the receptor. (Docking energy: -9.21 kcal/mol) |
[50] |
[16] Wu, Q.; Zhang, Y.; Tang, H.; Chen, Y.; Xie, B.; Wang, C.; Sun, Z. Separation and Identification of Anthocyanins Extracted from Blueberry Wine Lees and Pigment Binding Properties toward beta-Glucosidase. J. Agric. Food Chem. 2017, 65, 216-223.
[49] Mazewski, C.; Kim, M.S.; Gonzalez de Mejia, E. Anthocyanins, delphinidin-3-O-glucoside and cyanidin-3-O-glucoside, inhibit immune checkpoints in human colorectal cancer cells in vitro and in silico. Sci. rep. 2019, 9, 11560.
[50] Sari, D.R.T.; Cairns, J.R.K.; Safitri, A.; Fatchiyah, F. Virtual Prediction of the Delphinidin-3-O-glucoside and Peonidin-3-O-glucoside as Anti-inflammatory of TNF-α Signaling. Acta Informatica Medica 2019, 27, 152.
[51] Huang, W.-Y.; Wang, X.-N.; Wang, J.; Sui, Z.-Q. Malvidin and its glycosides from vaccinium ashei improve endothelial function by anti-inflammatory and angiotensin I-converting enzyme inhibitory effects. Nat. Prod. Commun. 2018, 13, 49-52.
[52] Srovnalova, A.; Svecarova, M.; Kopecna Zapletalova, M.; Anzenbacher, P.; Bachleda, P.; Anzenbacherova, E.; Dvorak, Z. Effects of anthocyanidins and anthocyanins on the expression and catalytic activities of CYP2A6, CYP2B6, CYP2C9, and CYP3A4 in primary human hepatocytes and human liver microsomes. J. Agric. Food Chem. 2014, 62, 789-797.
C6. Strengthen the link between findings and potential industrial applications.
(Author’s response)
Thank you for your comment. We agree with your opinion and added an explanation to provide a broader discussion. (Lines 345-349)
(After) These detailed finding will be suggested as fundamental data for the development of Korean superior cultivars and functional products, conditional variation evaluation, related field such as clinical and metabolomic research as well as the further research of anthocyanin variations by harvest time, cultivated methods and storage conditions.
C7. Comment on future studies on environmental or storage conditions affecting acetylated anthocyanin stability.
(Author’s response)
Thank you for your comment. We agree with your opinion and added an explanation to provide a broader discussion. (Lines 292-299)
(After) The synthesis of acylated anthocyanins can be catalyzed by acyltransferases, but the factors affecting the acetylation pathway are still unclear. Thus, further studies are required to elucidate the enzymatic activity and genetic regulation mechanism involved in the synthesis. In addition, the bioavailability, variations metabolic processes, and biological activities of anthocyanins, including acetylated glycosides should be evaluated in the future, and factors such as pH, light, and humidity need to investigated to assess their stability and expand their applications.

Reviewer 3 Report
Comments and Suggestions for Authors
The authors tested a series of flavonoids in multiple blueberries by using LC-MS/MS. The depth and significance are to be enhanced. And the listed issues should be noted.
Table 1, the harvest season column is very unclear. What does "season" mean, like summer and winter? If the season factor is quite important, please make it clear, not simply "early" or "mid"; if it is not so important, please discard this column since it is not important. Besides, the variety of the samples seems narrow, either early or mid, either northern highbush or southern highbush. The key distinct factor among these cultivars should be expressed.
Line 135, there is no linear regression information shown in Table 2.
I suggest using "flavonoid glycosides" instead of "anthocyanins" or "anthocyanin glycosides".
Figure 3. The upper panel and the lower panel are not using the same scale on the x-axis. Besides, what is the ISTD "delphin"? And it is not defined in the Method section. The peak shape of ISTD is quite different from others, why? And what is it used for? According to the description, it seems to be an external standard method for quantitation, therefore the ISTD is used for just calibrating the RT, is it true?
Line 196. The discussion of elution order is not interesting. Identification can be done clearly by using LC-MS/MS, right? Galactoside and arabinoside are different in molecular weight. The authors can provide MS fragmentation in Figure for every compound, together with their spectra.
Line 225. According to Table 3, only 11 compounds were tested for method validation. But here the authors claimed "the composition and content of 22 anthocyanins", which is not scientific. The work should be "semi-quantitative" for these flavonoids. Why are peaks #2, #3, #5~9, and #11~14 annotated as "External standard"?
Line 250. There is no color-related data shown in this manuscript, so, the discussion seems lonely. Similarly, the "synthesis of acylated anthocyanins" cannot be discussed deeply.
Since the authors tested so many flavonoids in multiple samples, I suggest using multivariate statistical analysis (e.g., PCA, HCA, OPLS-DA) to get further results regarding the profiling, the characteristics, etc.
Line 284, Since the sample size is not so large, I suggest the authors not give too dicitive conclusions.
Author Response
Response to reviewer's comments
Thanks very much for the reviewer's comments and kind advice on our manuscript. All of these comments have helped us so much to improve the manuscript. We have studied these comments carefully, and our responses to all the comments are given below in a way of point by point. Besides, some changes have been made throughout the whole manuscript, and all the modifications, corrections and changes are highlighted by red color in the revised manuscript.
C1. Table 1, the harvest season column is very unclear. What does "season" mean, like summer and winter? If the season factor is quite important, please make it clear, not simply "early" or "mid"; if it is not so important, please discard this column since it is not important. Besides, the variety of the samples seems narrow, either early or mid, either northern highbush or southern highbush. The key distinct factor among these cultivars should be expressed.
(Author’s response)
Thank you for your comments. We agree with the reviewer’s comment that the harvest season is unclear. Therefore, we have added the harvest season content to lines 41-43 as suggested and revised Table 1.
(After) Early-season cultivars are typically harvested from late May to early June, while mid-season cultivars are harvested from mid- to late June, and late-season cultivars are harvested from early to late July [4].
[4] Kim SJ, Jeong SM, Heo YY, Nam JC, Kim SH, Cho KH, Park SJ. Analysis of growth period and morphological characteristics of introduced blueberry cultivars in Korea. Korean J Plant Res. 2017,30, 101-109.
Cultivar |
Harvest season |
Reka |
Early |
Hannah’s choice |
Early |
Spartan |
Early |
Draper |
Early |
Patriot |
Early |
Legacy |
Mid |
Suziblue |
Early |
Farthing |
Early |
Newhanover |
Mid |
C2. Line 135, there is no linear regression information shown in Table 2.
(Author’s response)
Thank you for your comment. We confirmed that the linear regression information is included in Table 3.
C3. I suggest using "flavonoid glycosides" instead of "anthocyanins" or "anthocyanin glycosides".
(Author’s response)
Thank you for your suggestion. "flavonoid glycosides" is a broader term that encompasses compounds and anthocyanins are a subset of flavonoids. The focus of our study is on anthocyanins and their glycosylated forms. We did not investigate other flavonoid compounds such as quercetin or kaempferol in this research. For this reason, we have retained the terms "anthocyanins" and "anthocyanin glycosides" to reflect the precise scope of the study.
C4. Figure 3. The upper panel and the lower panel are not using the same scale on the x-axis. Besides, what is the ISTD "delphin"? And it is not defined in the Method section. The peak shape of ISTD is quite different from others, why? And what is it used for? According to the description, it seems to be an external standard method for quantitation, therefore the ISTD is used for just calibrating the RT, is it true?
(Author’s response)
Thank you for your comments regarding Figure 3 and the use of the ISTD "delphin." We explain your points below:
- Different x-axis scales in the upper and lower panels of Figure 3: We agree with the reviewer’s comment that not using the same scale on the x-axis. Therefore, we have revised Figure 3.
(After)
Figure 3. MRM chromatograms of 11 mixed standards (a) and UPLC-DAD chromatograms of 22 anthocyanins in Patriot (wavelength at 520 nm) (b). IS (internal standard): delphin 100 ppm
- Definition and Purpose of the ISTD Delphin: We were unable to obtain authentic standards for all target compounds, making the use of an internal standard unavoidable. For compounds with available standards, external quantification was performed. Therefore, the author revised and added related information in the materials and methods (line 132-136) and references have been added to support the justification.
(After) Material and Methods: Through preliminary experiments, delphin, which does not overlap with sample peaks and has a similar structure to blueberry anthocyanins, was selected as ISTD, and the relative quantification was calculated by comparing the relative peak areas of the compounds (based on major fragment ions) and ISTD on a 1:1 basis without considering the relative response factor. The external quantification was performed in MRM mode with selected 11 standards.
Lines 132-136
Heinle, L., Sulaiman, K., Olson, A., & Ruterbories, K. (2020). A homologous series of internal standards for near universal application in the discovery LC-MS/MS bioanalytical laboratory. Journal of Pharmaceutical and Biomedical Analysis, 190, 113578.
Lee, S.; Kwon, R.H.; Kim, J.H.; Na, H.; Lee, S.-J.; Choi, Y.-M.; Yoon, H.; Kim, S.Y.; Kim, Y.-S.; Lee, S.H. Changes in isoflavone profile from soybean seeds during Cheonggukjang fermentation based on high-resolution UPLC-DAD-QToF/MS: New succinylated and phosphorylated conjugates. Molecules 2022, 27, 4120.
Ribas-Agustí, A., Cáceres, R., Gratacós-Cubarsí, M., Sárraga, C., & Castellari, M. (2012). A validated HPLC-DAD method for routine determination of ten phenolic compounds in tomato fruits. Food Analytical Methods, 5, 1137-1144.
- Peak Shape of ISTD Compared to other Compounds: This phenomenon can be mainly attributed to the mobile phase composition, column selection, and analysis conditions. In particular, we used 5% formic acid in water and a water/acetonitrile (1:1, v/v) with 5% formic acid as the mobile phase. The analytical conditions using such solvents were optimized considering the chemical properties of anthocyanins and were suitable for the separation of anthocyanins. In addition, anthocyanins with a structure of two sugars, such as delphinidin 3,5-di-O-glucoside (delphin), is likely to be eluted relatively in the front depending on the mobile phase composition and column conditions, and it is suggested that the combination of sugars increases the interaction between molecules or changes the affinity for the mobile phase, resulting in a broad shape.
Choung, M. G. (2008). Optimal HPLC condition for simultaneous determination of anthocyanins in black soybean seed coats. Korean Journal of Crop Science, 53(4), 359-368.
C5. Line 196. The discussion of elution order is not interesting. Identification can be done clearly by using LC-MS/MS, right? Galactoside and arabinoside are different in molecular weight. The authors can provide MS fragmentation in Figure for every compound, together with their spectra.
(Author’s response)
Thank you for your comment. galactosides and arabinosides can be easily distinguished due to their different molecular weights. However, for derivatives with similar molecular weights and MS patterns, such as galactose and glucose, elution order can serve as a basis for comparison. Therefore, discussing the elution order is important, and MS spectral data has also been provided Figure S1.
(After)
Figure S1. Fragmentation (m/z, [M]+) patterns of anthocyanins identified in highbush blueberry (a, peak 2) delphinidin 3-O-glucoside, (b, peak 3) cyanidin 3-O-galactoside, (c, peak 4) delphinidin 3-O-arabinoside, (d, peak 5) cyanidin 3-O-glucoside, (e, peak 6) petunidin 3-O-galactoside, (f, peak 7) cyanidin 3-O-arabinoside, (g, peak 8) petunidin 3-O-glucoside, (h, peak 9) peonidin 3-O-galactoside, (i, peak 10) petunidin 3-O-arabinoside, (j, peak 11) peonidin 3-O-glucoside, (k, peak 13) peonidin 3-O-arabinoside, (l, peak 14) malvidin 3-O-glucoside, (m, peak 15) malvidin 3-O-arabinoside.
C6. Line 225. According to Table 3, only 11 compounds were tested for method validation. But here the authors claimed "the composition and content of 22 anthocyanins", which is not scientific. The work should be "semi-quantitative" for these flavonoids. Why are peaks #2, #3, #5~9, and #11~14 annotated as "External standard"?
(Author’s response)
Thank you for your comment. We have carefully reviewed our methodology and confirm that our analysis is based on quantitative methods rather than semi-quantitative methods. Peaks #2, #3, #5–9, and #11–14 were labeled as "External standard" as they were quantified they were quantified using external standards we possess. However, based on your feedback, we understand that the use of internal standards for some compounds might cause confusion about semi-quantitative aspects. Thus, we added a footnote about the use of internal standards for these compounds. lines 257-263
(After) 1) 1, delphinidin 3-O-galactoside; 2, delphinidin 3-O-glucoside; 3, cyanidin 3-O-galactoside; 4, delphinidin 3-O-arabinoside; 5. cyanidin 3-O-glucoside; 6, petunidin 3-O-galactoside; 7, cyanidin 3-O-arabinoside; 8, petunidin 3-O-glucoside; 9, peonidin 3-O-galactoside; 10, petunidin 3-O-arabinoside; 11, peonidin 3-O-glucoside; 12, malvidin 3-O-galactoside; 13, peonidin 3-O-arabinoside; 14, malvidin 3-O-glucoside; 15, malvidin 3-O-arabinoside; 16, delphinidin 3-O-(6''-O-acetyl)glucoside; 17, petunidin 3-O-(6''-O-acetyl)galactoside; 18, cyanidin 3-O-(6''-O-acetyl)glucoside; 19, petunidin 3-O-(6''-O-acetyl)glucoside; 20, malvidin 3-O-(6''-O-acetyl)galactoside; 21, peonidin 3-O-(6''-O-acetyl)glucoside; 22, malvidin 3-O-(6''-O-acetyl)glucoside. Each value calculated as means ± SD (n = 3) using an internal standard (delphin) and *External standards. a-fDifferent superscript letters next to mean values (n = 3) indicate significant differences (p < 0.05) according to Duncan’s multiple range test.
C7. Line 250. There is no color-related data shown in this manuscript, so, the discussion seems lonely. Similarly, the "synthesis of acylated anthocyanins" cannot be discussed deeply.
(Author’s response)
Thank you for your feedback. We agree with the reviewer's comment that color-related data were not presented in this manuscript. Therefore, we have moved the discussion of color-related properties of anthocyanins and their derivatives from the Conclusion to the discussion of acetylanthocyanins to the Conclusion lines 284-299.
(After)
Conclusion: The seven acetylated glycosides (peaks 16–22) accounted for 19.7% of total anthocyanin were highest in ‘Patriot’ (179.5), whereas these glycosides were not detected in Legacy, Draper, and Farthing (Table 5). Acetylated glycosides of ‘Patriot’ showed significant proportions of delphinidin and malvidin derivatives, which were consistent with data reported for Canadian lowbush and highbush blueberries [45]. Acetylated anthocyanins may have metabolic health benefits, including improved insulin sensitivity, reduced inflammation, and regulation of gut microbiota [46]. Malvidin 3-O-(6"-O-acetyl)glucoside has demonstrated free radical scavenging activity [47] and potential anti-inflammatory effects via selective COX-2 and β-glucosidase inhibition [48,16]. The synthesis of acylated anthocyanins can be catalyzed by acyltransferases, but the factors affecting the acetylation pathway are still unclear. Thus, further studies are required to elucidate the enzymatic activity and genetic regulation mechanism involved in the synthesis. In addition, the bioavailability, variations metabolic processes, and biological activities of anthocyanins, including acetylated glycosides should be evaluated in the future, and factors such as pH, light, and humidity need to investigated to assess their stability and expand their applications.
C8. Since the authors tested so many flavonoids in multiple samples, I suggest using multivariate statistical analysis (e.g., PCA, HCA, OPLS-DA) to get further results regarding the profiling, the characteristics, etc.
(Author’s response)
Thank you for your comment. We agree with the reviewer's comment that the statistical analysis was insufficient. Therefore, we have added related information as suggested lines 155-167, lines 306-329, lines 342-345 and additionally presented Figure 5, Figure S1, Figure S2 and Table S1.
(After)
Thank you for your comment. We agree with the reviewer's comment that the statistical analysis was insufficient. Therefore, we have added relevant information as suggested abstract (ines 24-27), introduction (lines 79-80), materials and methods (ines 162-175), results and discussion (lines 300-329), conclusions (lines 340-343) and additionally presented Figure 5, Figure S2, Figure S3 and Table S1.
(After)
Abstract: Multivariate analysis showed a distinct separation between early- and mid-seasons with Draper. Especially, delphinidin 3-O-galactoside (VIP = 1.94) was identified as a marker for mid-season, malvidin 3-O-glucoside (VIP = 1.79) was identified as a marker for early-season. (lines 24-27)
Introduction: In addition, multivariate analysis visualized differences between cultivars and identified discriminant markers critical for cultivar differentiation. (lines 79-80)
Method: 2.5. Muliple anlysis for anthocyanins
A multivariate analysis was conducted using SIMCA (version 16, Satorius Stedim Data Analysis AB, Sweden). Principal component analysis (PCA) and orthogonal partial least squares discriminant analysis (OPLS-DA) were applied for statistical comparison and marker identification of nine blueberry cultivars. The scaling method was performed using Pareto (par). Chemical markers that distinguish the two species were obtained by combining S- and variable importance in projection (VIP) plots. Chemical markers were selected with VIP > 1 and p-value < 0.05 to indicate their influence on classification. To evaluate the reliability and predictability of the OPLS-DA model, 7-fold cross validation (CV) and 200 random permutation tests, and receiver operating characteristic (ROC) curve models were performed. Additionally, hierarchical clustering analysis (HCA) with a heat map was constructed using Euclidean distance method using MetaboAnalyst online analysis software (http://www.metaboanalyst.ca) for clustering and visualization of the chemical markers between early and mid-seasons. (lines 162-175)
Results and Dis:cussion: 3.3. Multivariate statistical analysis of highbush blueberry cultivars
Multivariate statistical analysis was performed to visualize the differences of anthocyanin between highbush blueberry cultivars and to identify chemical markers contributing to these differences. The PCA score plot (R2X = 0.976, Q2 = 0.887) revealed three clusters (cluster 1, New hanover; cluster 2, Draper, Legacy, Reka, Spartan, Suziblue and Hannah's choice; cluster 3, Patriot). Patriot showed distinct differences from other cultivars, primarily attributed to its high contents of acetyl glycosides. As early-season, Draper was found to share more similar compounds with mid-season (Figure 5a). The PCA loading plot demonstrated variables corresponding to each cultivar (Figure S1a). To further clarify the separation between two groups (early vs. mid) and identify differential markers, OPLS-DA analysis was additionally conducted (R2X = 0.937, R2Y = 0.819, Q2 = 0.644). The OPLS analysis formed three clusters, with Draper positioned relatively close to the mid-season group, similar to the PCA results (Figure 5b). The OPLS loading plot highlighted variables with VIP > 1 in red, showing that compounds predominantly found in early-season were distributed in the second quadrant, and those in mid-season were in the fourth quadrant, and these variables were identified as significant components distinguishing the cultivars (Figure S1b). The accuracy and reliability of the 22 markers were validated through the AUCs and permutation plots for blueberries, indicating no overfitting in the model (Figure S2). The CV-ANOVA analysis yielded a p-value of 0.028 (p < 0.05), confirming that the established OPLS-DA model is statistically significant and reliable. The S-plot visualized the variables that distinguish the two groups, especially, delphinidin 3-O-galactoside (VIP = 1.94) was identified as a marker for mid-season, while malvidin 3-O-glucoside (VIP = 1.79) was identified as a marker for early-season (Figure 5c, Table S1). HCA was also performed to visualize the differences in content between early and mid-seasons for variables with VIP > 1, and similar to the OPLS-DA results, it distinguished two groups (early vs. mid). In conclusion, five markers (delphinidin 3-O-galactoside, delphinidin 3-O-arabinoside, petunidin 3-O-galactoside, malvidin 3-O-galactoside and malvidin 3-O-arabinoside) of mid-seasons with Draper and three markers (delphinidin 3-O-glucoside, petunidin 3-O-glucoside and malvidin 3-O-glucoside) of early seasons were confirmed (Figure 5d). (lines 300-329)
Conclusion: Multivariate analysis showed clear separation between early- and mid-seasons with Draper. Among the 8 markers, delphinidin 3-O-galactoside (VIP = 1.94) was identified as a marker for mid-season, malvidin 3-O-glucoside (VIP = 1.79) was identified as a marker for early-season. (lines 340-343)
Figure 5. The multivariate statistical analysis of 9 highbush blueberry cultivars. (a) PCA score plot, (b) OPLS-DA score pot, (c) OPLS-DA S-plot, (d), and heat map of HCA.
Figure S2. Validation of the OPLS-DA model for highbush blueberry cultivars (a) ROC curve, (b) permutation test for early-season and (c) permutation test for mid-season.
Figure S3. Multivariate analyses in 9 highbush blueberry cultivars based on MS data. (a) PCA loading plot and (b) OPLS-DA loading plot.
Table S1. VIP values and p-values of markers identified between early and mid-seasons by OPLS-DA and t-test
Compound |
1)VIP-value |
2)p-value |
Delphinidin 3-O-galactoside |
1.94 |
1.98E-05 |
Delphinidin 3-O-glucoside |
1.84 |
4.06E-02 |
Delphinidin 3-O-arabinoside |
1.79 |
5.61E-04 |
Petunidin 3-O-galactoside |
1.61 |
5.67E-05 |
Petunidin 3-O-glucoside |
1.29 |
1.86E-03 |
Malvidin 3-O-galactoside |
1.08 |
4.39E-05 |
Malvidin 3-O-glucoside |
1.08 |
6.05E-04 |
Malvidin 3-O- arabinoside |
1.02 |
3.64E-02 |
1) VIP was obtained from OPLS-DA with a threshold of 1.
2) p-value was calculated with t-test
C9. Line 284, Since the sample size is not so large, I suggest the authors not give too dicitive conclusions.
(Author’s response)
Thank you for your comment. Based on your suggestion, we have revised the conclusion to reflect the opinion that the sample size is not large and to avoid definitive statements. (Lines 345-349)
(After)
Conclusion: These detailed finding will be suggested as fundamental data for the development of Korean superior cultivars and functional products, conditional variation evaluation, related field such as clinical and metabolomic research as well as the further research of anthocyanin variations by harvest time, cultivated methods and storage conditions. (Lines 345-349)

Round 2
Reviewer 1 Report
Comments and Suggestions for Authors
The Authors have been positively answered to my remarks therefore the manuscript has been improved. I recommend to publish manuscript in present form
Reviewer 3 Report
Comments and Suggestions for Authors
The authors did modifications and revisions according to my comments, and improved the overall quality of this manuscript. Although there are still some differences of opinion between the reviewers and the authors on certain issues, such as methodology, these do not affect the publishability of the article. Therefore, I agree to accept the manuscript for publication.